# Immune Landscape in PTEN-Related Glioma Microenvironment: A Bioinformatic Analysis

**DOI:** 10.3390/brainsci12040501

**Published:** 2022-04-14

**Authors:** Alice Giotta Lucifero, Sabino Luzzi

**Affiliations:** 1Neurosurgery Unit, Department of Clinical-Surgical, Diagnostic and Pediatric Sciences, University of Pavia, 27100 Pavia, Italy; alicelucifero@gmail.com; 2Neurosurgery Unit, Department of Surgical Sciences, Fondazione IRCCS Policlinico San Matteo, 27100 Pavia, Italy

**Keywords:** glioblastoma, high-grade gliomas, phosphatase and tensin homolog, PTEN, temozolomide, The Cancer Genome Atlas

## Abstract

Introduction: PTEN gene mutations are frequently found in the genetic landscape of high-grade gliomas since they influence cell proliferation, proangiogenetic pathways, and antitumoral immune response. The present bioinformatics analysis explores the PTEN gene expression profile in HGGs as a prognostic factor for survival, especially focusing on the related immune microenvironment. The effects of PTEN mutation on the susceptibility to conventional chemotherapy were also investigated. Methods: Clinical and genetic data of GBMs and normal tissue samples were acquired from The Cancer Genome Atlas (TCGA)-GBM and Genotype-Tissue Expression (GTEx) online databases, respectively. The genetic differential expressions were analyzed in both groups via the one-way ANOVA test. Kaplan–Meier survival curves were applied to estimate the overall survival (OS) and disease-free survival (DFS). The Genomics of Drug Sensitivity in Cancer platform was chosen to assess the response of PTEN-mutated GBMs to temozolomide (TMZ). *p* < 0.05 was fixed as statistically significant. On Tumor Immune Estimation Resource and Gene Expression Profiling Interactive Analysis databases, the linkage between immune cell recruitment and PTEN status was assessed through Spearman’s correlation analysis. Results: PTEN was found mutated in 22.2% of the 617 TCGA-GBMs patients, with a higher log2-transcriptome per million reads compared to the GTEx group (255 samples). Survival curves revealed a worse OS and DFS, albeit not significant, for the high-PTEN profile GBMs. Spearman’s analysis of immune cells demonstrated a strong positive correlation between the PTEN status and infiltration of T_reg_ (ρ = 0.179) and M2 macrophages (ρ = 0.303). The half-maximal inhibitor concentration of TMZ was proven to be lower for PTEN-mutated GBMs compared with PTEN wild-types. Conclusions: PTEN gene mutations prevail in GBMs and are strongly related to poor prognosis and least survival. The infiltrating immune lymphocytes T_reg_ and M2 macrophages populate the glioma microenvironment and control the mechanisms of tumor progression, immune escape, and sensitivity to standard chemotherapy. Broader studies are required to confirm these findings and turn them into new therapeutic perspectives.

## 1. Introduction

High-grade gliomas (HGGs) are common neoplasms of the central nervous system accounting for 70% of brain tumors [1,2,3,4,5]. Glioblastoma (GBM) is the most lethal and represents 60% of newly diagnosed gliomas in the adult population [6,7]. The current standard of care in the management of HGGs is maximum surgical resection, adjuvant chemoradiation, and six cycles of temozolomide [8].

Despite advances in surgical techniques, diagnostics, and target therapeutic strategies, the 5-year survival rate persists under 10% and the median overall survival (OS) still ranges between 14 and 16 months [9]. The poor prognosis and high mortality rate of GBM are attributable to aberrant angiogenesis, extreme mitotic activity, immune escape mechanisms, and intrinsic genome-wide heterogeneity [10,11,12,13,14,15,16,17]. In 2021, Louis and colleagues published the fifth edition of the WHO classification of brain cancers, which reflects the advances in translational medicine, taxonomy, and genetics in neuro-oncology.

They reported a novel tumors nomenclature aimed at integrating histological features, key diagnostic genes, and molecular characteristics underlying oncogenesis [18].

Phosphatase and tensin homolog (PTEN), a tumor suppressor gene, is closely involved in cell translation, proliferation, and tumorigenesis [19,20,21,22]. PTEN protein blocks the intracellular pathways of phosphatidylinositol 3-kinase/AKT/mammalian target of rapamycin (PI3K/AKT/mTOR) via dephosphorylation of phosphatidylinositol-3-triphosphate, resulting in inhibition of the cell cycle [23,24,25,26].

Moreover, the PTEN pathway has a close interaction with the Wnt/β-catenin signals involved in embryogenesis and the determination of neural patterning [22].

Lack/mutation of the PTEN gene, found in 40% of GBMs, influences neurogenesis, and gliogenesis and heightens the DNA damage repairing and the malignant progression of brain tumors [20,25,27].

The prognostic significance of the PTEN gene is also related to the maintenance of the immune microenvironment [28]. Recent transcriptomic pieces of evidence support the correlation between PTEN mutation and the amendment of immune infiltrating cells expression [29,30]. The immune suppression mechanisms intrude on the host antitumor responses and are liable for the failure of conventional therapeutic approaches [31,32,33,34,35]. The number of reported studies on the immunogenomics and immunosuppressive microenvironment of HGGS opened the way for tailored treatments against glioma resilience [36,37,38,39]. Based on these assumptions, we conducted a bioinformatics analysis to examine the PTEN gene expression profile in HGGs as a prognostic factor for survival. We investigated the cluster of the immune infiltrates within the PTEN-related microenvironment intending to identify the contribution of each subpopulation to the immune escape mechanisms. The effects of PTEN mutation on the sensitivity to standard chemotherapy were also explored.

## 2. Materials and Methods

### 2.1. Data Acquisition

Transcriptomes, genetic, and clinical features of HGGs-patients were extracted from The Cancer Genome Atlas (TCGA)-GBM project (https://portal.gdc.cancer.gov) (accessed on 31 October 2021) [40]. The genetic data of normal brain tissue samples were acquired from the Genotype-Tissue Expression (GTEx) online database (https://gtexportal.org) (accessed on 1 November 2021).

### 2.2. Statistical Bioinformatics Analysis

R (https://www.r-project.org) (accessed on 7 March 2022) and Prism 5 (GraphPad Software, Inc., La Jolla, CA, USA) software were used for statistical analysis. Continuous and categorical variables were reported as mean and percentages, respectively.

Top mutation trends and gene patterns were estimated in the TCGA-GBM cohort, focusing on nucleotide variations of the PTEN gene. The differential expressions of PTEN mRNA levels were determined in TCGA-GBM and GTEx groups with the aim to assess the significance of the PTEN gene in the glioma genome compared to healthy brain tissue. A one-way ANOVA test was used for the analysis.

Kaplan–Meier survival curves were used to estimate the prognostic value of the PTEN gene mutations. The overall survival (OS), disease-free survival (DFS), and comparisons between the high- or low-PTEN mutation profile in TCGA-GBM patients were assessed using the log-rank test. Hazard ratios (HDs) were calculated with the Cox proportional risk regression model. The Genomics of Drug Sensitivity in Cancer (GDSC) database (https://www.cancerrxgene.org/) (accessed on 15 November 2021) was the source used for the appraisal of the chemotherapeutic response of PTEN-mutated HGGs [41]. The R software package “Prophetic” was used and the half-maximal inhibitor concentration (IC50) of temozolomide (TMZ) was assessed by one-way ANOVA analysis [41]. *p*-value was set at <0.05 for all the tests.

### 2.3. Estimation of Immune Infiltrating Cells

On behalf of the glioma immune microenvironment, the T cells CD4+, CD8+, T_reg_, NK cells, monocytes/macrophages, and tumor-infiltrating endothelial cells were considered in the TCGA-GBM cohort. Tumor Immune Estimation Resource 2.0 (TIMER2.0) was used to identify the correlation between mRNA PTEN expression and the transcriptional profile of each tumor immune cell in the TCGA-GBM project.

Assuming the purity adjustment, Spearman’s correlation method was employed, where rho (ρ) > 0 and ρ < 0 denoted a positive and negative correlation between the variables, respectively. Based on the Gene Expression Profiling Interactive Analysis (GEPIA), the differential subexpression of mRNA PTEN in the immune subtypes, for both TCGA-GBM and GTEx samples, was clustered with the one-way ANOVA method, and *p* < 0.05 was assumed as statistical. The results were reported as boxplots.

## 3. Results

### 3.1. Demographics and Gene Mutation Profiles

The genetic and clinical data of 617 HGGs were collected by the TCGA-GBM project. The average patients’ age was 58.8 ± 14 years; males were 59.5%, and 96.3% were Caucasian. All the tumors were supratentorial glioblastoma. Chemotherapy was administered as adjuvant treatment in 52.2% of patients, while 45% were treated with concomitant radiotherapy. The average follow-up was 14.7 months, and 69.2% of patients were dead. Overall data about TCGA-GBM patients are summarized in Table 1.

PTEN was the most frequent gene by the transcriptome examination (22.2%). It was followed by TTN (20.75%), TP53 (20.10%), EGFR (17.18%), FLG (12.64%), MUC16 (11.51%), NF1 (8.27%), RYR2 (7.62%), PKHD1 (7.29%), HMCN1 (7.29%), SYNE1 (7.29%), SPTA1 (6.97%), PIK3R1 (6.97%), RB1 (6.81%), ATRX (6.65%), IDH (6.62%), PIK3CA (6.48%), OBSCN (6.48%), APOB (6.32%), FLG2 (6.32%), and LRP2 (6.16%) (Figure 1).

Expression profiles and mutations of the top 50 genes in TCGA-GBM are shown in the Oncogrid (Figure 2).

From the GTEx dataset, a total of 255 samples of normal brain tissue were included. The differential analysis revealed a higher expression of PTEN mRNA levels in the tumor than in normal tissue, albeit not significant, with a log2-transcriptome per million reads (TPM) +1 of 2.8–5.75 and 2.7–4.8 in the TCGA-GBM and GTEx datasets, respectively (Figure 3).

### 3.2. Survival Analysis

Kaplan–Meier curves showed a worse, though non statistic, OS (Log-rank *p* = 0.58, HR = 0.91, *p* = 0.6) and DFS (Log-rank *p* = 0.78, HR = 1, *p* = 0.82) in high-PTEN mutation profile (Figure 4).

### 3.3. Immune Landscape in PTEN-Related Glioma Microenvironment

Spearman’s correlation analysis of immune subpopulations applied in the TCGA-GBM cohort revealed a negative correlation between mRNA PTEN mutations expression (log2 TPM) and infiltrating T cell CD4+ Th 1 (ρ = −0.245; *p* = 3.81 × 10^−3^), Th 2 (ρ = −0.225, *p* = 8.06 × 10^−3^), and NK cells (ρ = −0.163, *p* = 5.64 × 10^−2^) within the tumor microenvironment. Conversely, T_reg_ (ρ = 0.179, *p* = 3.54 × 10^−2^), endothelial cells (ρ = 0.303, *p* = 2.97 × 10^−4^), and monocyte/macrophages (ρ = 0.368, *p* = 9.17 × 10^−6^) were predominant, with a polarization of M2 (ρ = 0.303, *p* = 2.97 × 10^−4^) against of the monocyte (ρ = −0.205, *p* = 1.6 × 10^−2^) (Figure 5 and Figure 6).

The GEPIA analysis of the mRNA PTEN mutations expression (logTPM +1) in the immune infiltrates of the TCGA-GBM cohort highlighted the decreased density of B cell naive (*p* < 1 × 10^−15^), T cell CD4+ naive (*p* < 1 × 10^−15^), T cell CD8+ (*p* = 0.08), and NK cell (*p* = 3.97 × 10^−9^) in the glioma microenvironment compared to GTEx samples. The T_reg_ subtype was slightly more represented (*p* = 0.07).

Furthermore, the boxplots evidenced the monocytes (*p* < 1 × 10^−15^), endothelial cells (*p* < 1 × 10^−15^), macrophages M0 (*p* = 3.49 × 10^−3^), M1 (*p* < 1 × 10^−15^), and M2 (*p* < 1 × 10^−15^) as the main components of the immune profile at the tumor site in the TCGA-GBM cohort (Figure 7 and Figure 8).

### 3.4. Prediction of Chemotherapeutic Response

Based on the GDSC pharmacogenomic database, lower differential targeted responses to TMZ were found in PTEN-mutated and PTEN wild-type samples, with a median IC50 of 531.13 and 701.55 μM, respectively (Figure 9).

## 4. Discussion

The present study analyzes mutation profiles and immune signatures of PTEN-associated microenvironment estimating GBM patients’ prognosis, survival, and chemotherapy response. The PTEN gene regulates the cell cycle and DNA repair mechanisms. Its expression modulates cell proliferation, neural development, and gliogenesis [21,27,42].

PTEN mutations are hallmarks of glioma malignancy and influence the patients’ survival [43,44,45,46,47,48,49]. As recently reported by Erira and his group, alterations of PTEN genes are related to glioma proliferation. PTEN mutation may induce post-translational changes in low-grade gliomas, leading to malignant progression [50].

The prognostic role of the PTEN status has been widely deepened in the literature. In 2001, Sasaki et al. analyzed the different median survival of glioma patients as it related to the PTEN expression. They reported a better OS for wild-type PTEN gliomas (123.4 months), compared to the mutated ones (14.8 months) [51]. In 2016, Han and colleagues explored the genetic linkage between PTEN expression and patients’ outcomes in a meta-analysis where a worse prognosis was revealed for PTEN-mutated gliomas [49]. Zhang and his group, 2021, conducted an online bioinformatics analysis about PTEN mutation as a prognostic signature for HGGs. They designed a tailored risk score based on the individual PTEN status, aiming to simplify HGGs diagnosis, prognosis, and treatment planning. They identified 14 independent prognostic genes in PTENwild-type tumors and 3 for the PTEN-mutated ones. These last proved to be related to the worse survival [52].

Even without any statistical significance, our Kaplan–Meier analysis confirmed dismal OS (Log-rank *p* = 0.58, HR = 0.91, *p* = 0.6) and DFS (Log-rank *p* = 0.78, HR = 1, *p* = 0.82) for the high-PTEN mutation profiles.

Amid the PTEN-related genetic mechanisms underlying the tumorigenesis, the maintenance of the glioma immune microenvironment is critical. Based on this rationale, our analysis also aimed at typifying the subpopulations in the glioma immune niche.

Our Spearman’s correlation tests of immune subpopulations reported a negative correlation (ρ < 0) between PTEN mutations and the expression of infiltrating T cell CD4+ Th 1 (ρ = −0.245), Th 2 (ρ = −0.225), and NK cells (ρ = −0.163) within the tumor microenvironment. The GEPIA one-way ANOVA analysis also revealed a lower density of B cell naive (*p* < 1 × 10^−15^), CD8+ (*p* = 0.08), CD4+ naive (*p* < 1 × 10^−15^), and NK cell (*p* = 3.97 × 10^−9^) concomitant with TCGA-GBM high-PTEN mutation profiles, compared to the GTEx samples.

In accordance with the evidence in the literature, these data denoted that brain tumor growth and progression are sustained by genetic mechanisms of immune tolerance and exhaustion. The suppression of T, B, and NK cells activity within the PTEN tumor microenvironment suggested immune-mediated biological processes are involved in pathways for glioma immune evasion and resistance to chemotherapies [53,54,55,56].

On the contrary, through Spearman’s analysis the T_reg_ (ρ = 0.179), endothelial cells (ρ = 0.303), and monocyte/macrophages (ρ = 0.368) were found to be prevalent. The differential examination demonstrated the polarization of M2 (ρ = 0.303) within the glioma microenvironment, versus the monocytes (ρ = −0.205). The GEPIA T_reg_ (*p* = 0.07) and macrophages (*p* = 3.49 × 10^−3^) were the most represented subtypes, with a prevalence of M2 in the TCGA-GBM group.

T_regs_ preserve immune homeostasis, contribute to the downregulation of T cell activity, and regulate innate and adaptive responses against self-antigens, allergens, and infectious agents [57,58,59,60,61]. T_regs_ also act as immunosuppressive within the tumor microenvironment, repressing the function of CD4+, CD8+, and NK cells [62]. The immune control is carried out by cytokines, extracellular vesicles, perforins, and cytolytic enzymes [63]. They repress antitumor immunity and facilitate immune escape mechanisms, resulting in glioma progression and relapse [64,65,66].

Our results also verified that the immunosuppressive anticancer microenvironment is sustained by the recruitment of monocytes, which in the glioma context are converted into macrophages, with an explicit M2 polarization. M2 macrophages are known to hold an immunosuppressive role [67,68]. M2 phenotypes induce the differential expression of receptors, cytokines, and chemokines. They produce IL-10, IL-1, and IL-6, stimulating tumorigenesis and negatively affecting the prognosis [69,70]. The M2 macrophages, detected in perivascular areas, enhance the VEGF and COX2 production resulting in increased and aberrant angiogenesis [71,72,73].

Therefore, T_reg_ and M2 cells stimulate the glioma cell proliferation, invasion, and support immune escape mechanisms [74,75,76,77]. These data were confirmed by our bioinformatic analysis (ρ = 0.3.03).

Opposing our results, the latest study by Zhou and colleagues published in 2022, found the high expression of PTEN related to a better prognosis for HGGs patients [78]. They discovered via the transwell and flow cytometry that the PTEN gene may inhibit the M1/M2 polarization and M2 macrophages recruitment. These data suggest a potential positive role of the PTEN as an antitumoral immunoregulatory gene [78].

The discrepancy in our study can be explained by the distinct patients cohorts involved, such as the Chinese Glioma Genome Atlas (CGGA) database, and the different techniques applied for data analysis. However, above all, they explored the effects of different PTEN statuses, split into PTEN deletion, PTEN mutated, and PTEN wild-type. This distinction allowed us to assess the specific impact of each PTEN gene expression on glioma immunity. The study demonstrates that, despite advances in genomics, further research is needed to shed light on PTEN activity and its immunological role in tumor progression.

The composition of immune infiltrates explains the prognosis concomitant to the PTEN status and, above all, the ineffective response to standard therapies. TMZ is currently the first line of treatment for HGGs in combination with surgery and radiotherapy [79,80]. It is still debated whether the PTEN mutation may influence sensitivity to radiochemotherapy [81,82,83]. PTEN controls the Wnt/β-catenin and PI3K/Akt/mTOR signaling pathways and arrests the cell cycle at the G2/M phase. TMZ alkylates DNA at this stage, hence PTEN overexpression affects the complex biochemical mechanism of drug alkylation encouraging TMZ activity [84,85].

In 2012 Carico and colleagues conducted a clinical study involving newly diagnosed GBM treated with TMZ. They reported greater effectiveness of TMZ in GBMs with PTEN loss. Inaba et al. investigated the effects of TMZ related to PTEN status founding an increased efficacy in cases of PTEN mutation [83]. Similarly, our analysis of the GDSC database revealed a lower IC50 for PTEN-mutated GBMs (531.13 μM) in comparison with wild-types (701.55 μM).

Apart from conventional chemotherapy, the identification of immune phenotypes and molecular interactions within the tumor microenvironment has been recognized as crucial to widening the spectrum of tailored strategies against the immune escape mechanisms [86,87,88].

### Limitations of the Study

The present study has several undeniable limitations, among which are the relatively limited number of patients and short follow-up (average 14.7 months). Other potential biases were the different patients’ ethnicity, limited data about the histological classification, and heterogeneity of radiochemotherapy regimens.

## 5. Conclusions

PTEN mutations frequently occur in malignant brain tumors, contributing to their progression, reduced OS, and DFS. Within the glioblastoma microenvironment, the PTEN-related immune landscape mainly consists of T_reg_ and M2 macrophages. They repress the antitumor immune activation and are responsible for the triggering of the glioma cell growth, invasion, and aberrant vasculogenesis.

PTEN expression and related glioma microenvironment also influence the sensitivity to conventional radiochemotherapy.

Prospective and randomized trials are necessary to validate these data and to develop novel target treatments.

## Figures and Tables

**Figure 1 brainsci-12-00501-f001:**
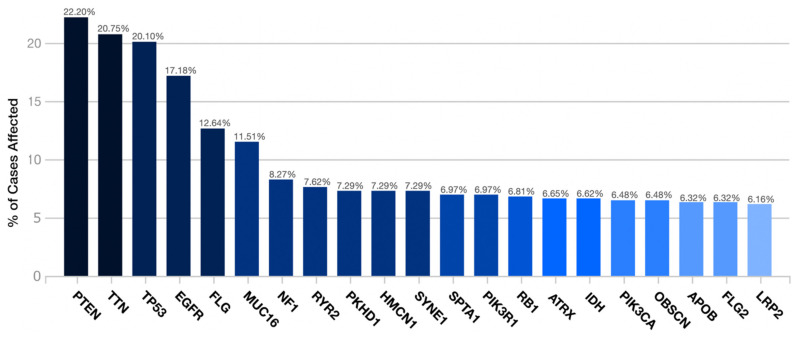
Distribution of the most frequent mutated genes in the TCGA-GBM project.

**Figure 2 brainsci-12-00501-f002:**
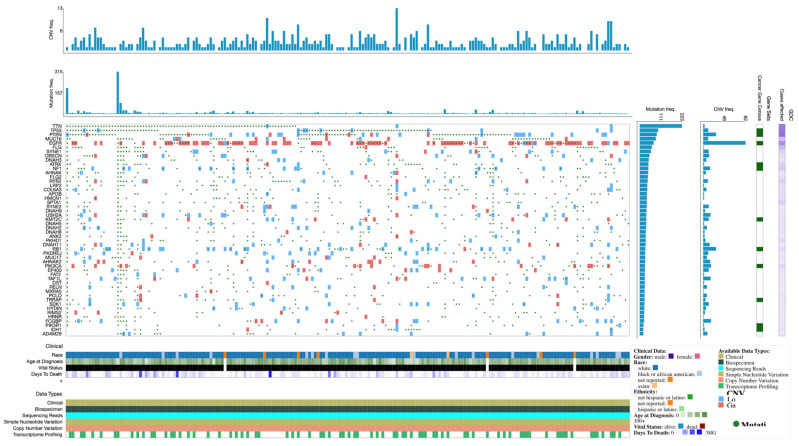
OncoGrid of top 50 mutated genes with impact mutations on the TCGC-GBM cohort.

**Figure 3 brainsci-12-00501-f003:**
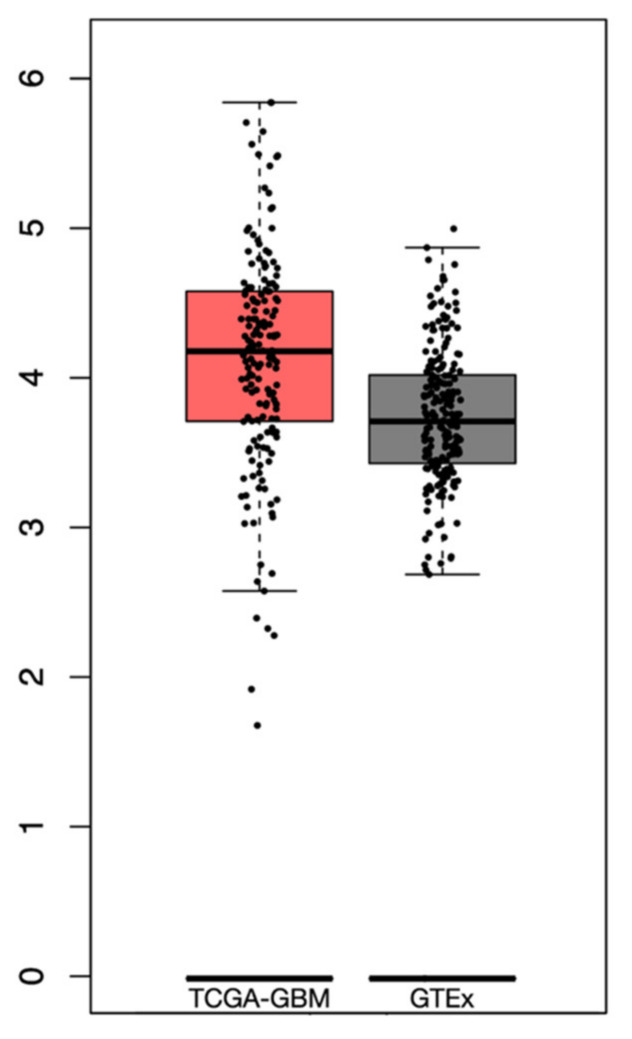
Box plots revealing the differential PTEN mRNA expression levels in TCGA-GBM and GTEx datasets.

**Figure 4 brainsci-12-00501-f004:**
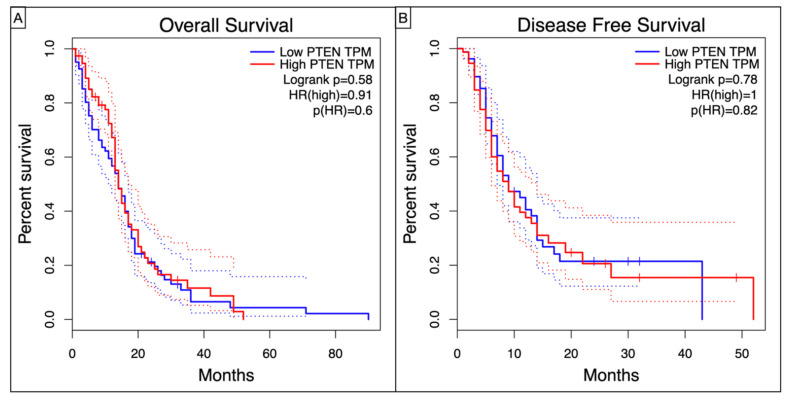
Kaplan–Meyer curves showing the (**A**) Overall Survival and (**B**) Disease Free Survival in TCGA-GBM patients according to the level of PTEN TPM.

**Figure 5 brainsci-12-00501-f005:**
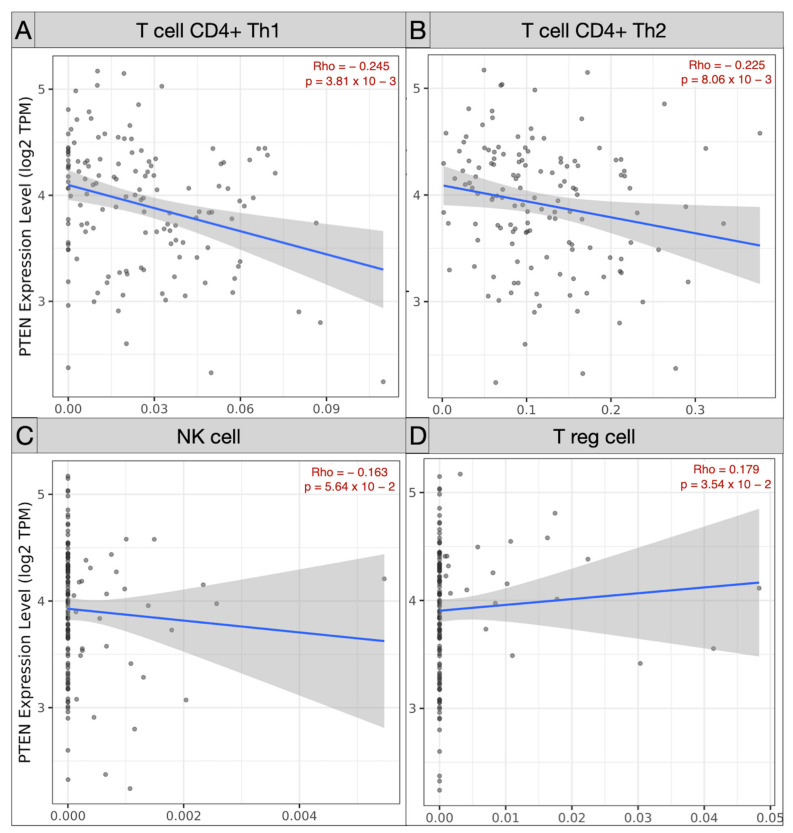
Scatter plots picturing the correlation of mRNA PTEN expression and the immune infiltration of T cell CD4 Th1 (**A**), Th2 (**B**), NK cell (**C**), and T_reg_ (**D**) in the TCGC-GBM project.

**Figure 6 brainsci-12-00501-f006:**
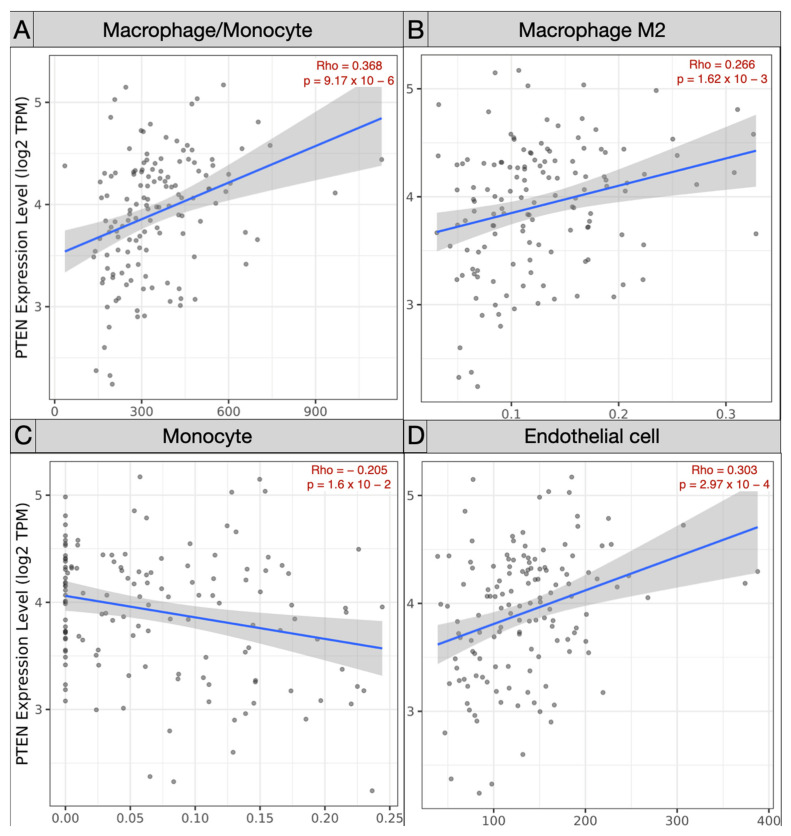
Scatter plots exhibiting the correlation of mRNA PTEN expression and the immune infiltration of (**A**) macrophage/monocyte, (**B**) macrophage M2, (**C**) monocyte, and (**D**) endothelial cells in the TCGA-GBM project.

**Figure 7 brainsci-12-00501-f007:**
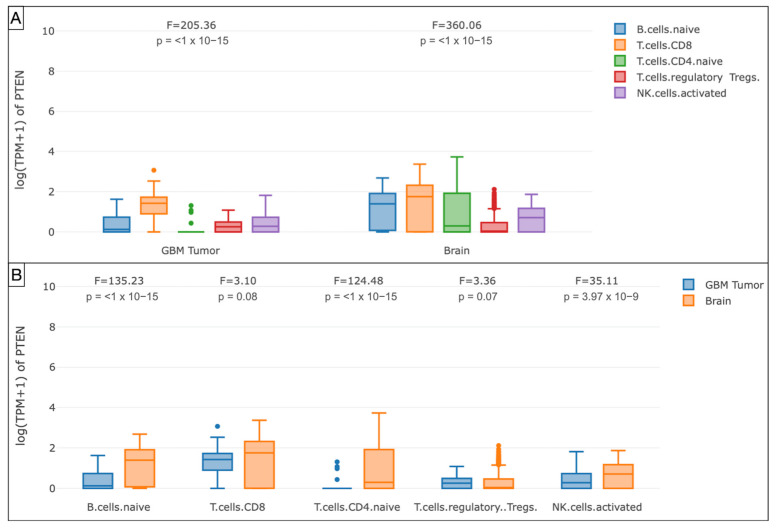
Box plots showing the immune cell subexpression analysis in PTEN-mutated TCGA-GBMs and normal brain samples from GTEx. (**A**) Grouped by tissue. (**B**) Grouped by B cells naive, T cells CD8, CD4, T_reg_, and NK cells.

**Figure 8 brainsci-12-00501-f008:**
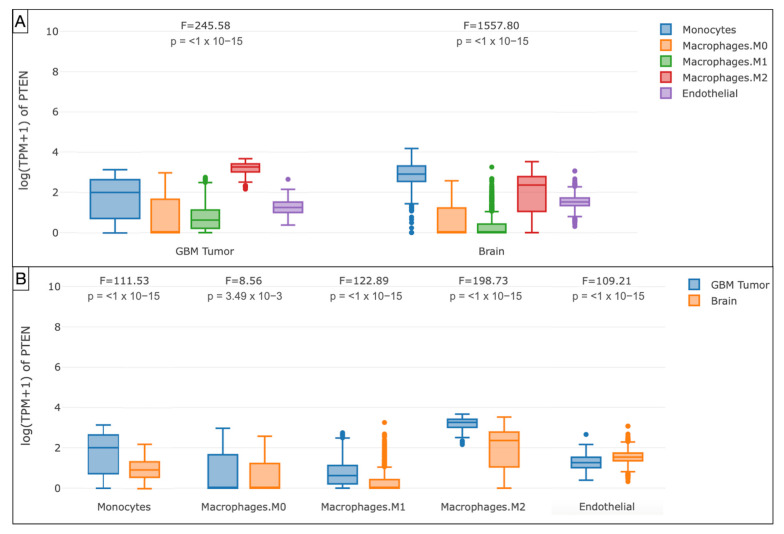
Box plots illustrating the immune cell subexpression analysis in PTEN-mutated TCGA-GBMs and normal brain samples from GTEx. (**A**) Grouped by tissue. (**B**) Grouped by monocytes, macrophages M0, M1, and M2, and endothelial cells.

**Figure 9 brainsci-12-00501-f009:**
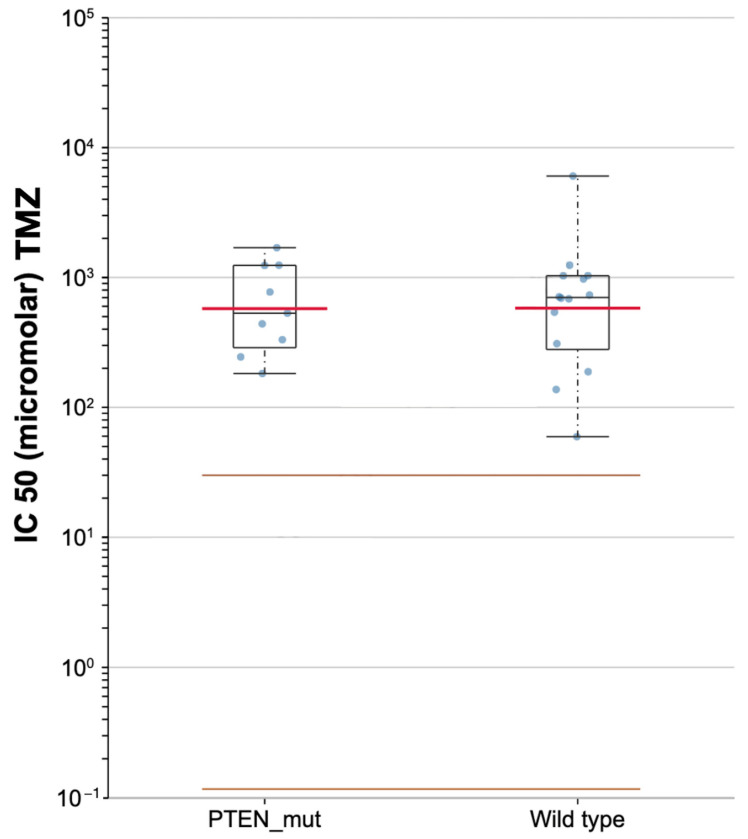
Scatter plot for the sensitivity to temozolomide (TMZ) in PTEN mutated and wild-type of TCGA-GBMs.

**Table 1 brainsci-12-00501-t001:** Demographic, clinical, and histological data of TCGA-GBM patients.

Variable	Data
Case no.	617
Average pts. age (yrs ± SD)	58.8 ± 14
Sex	
Male no. (%)	367 (59.5)
Female no. (%)	230 (37.3)
NOS no. (%)	20 (3.2)
Ethnicity	
Caucasian no. (%)	594 (96.3)
African no. (%)	20 (3.2)
Asian no. (%)	3 (0.5)
Histological type	
Glioblastoma no. (%)	617 (100)
IDH wild-type ratio/IDH-mutant ratio	577:40
Primary site	
Brain no. (%)	617 (100)
Adjuvant treatment	
Pharmacotherapy no. (%)	322 (52.2)
Radiation Therapy no. (%)	278 (45)
NOS no. (%)	17 (2.8)
Vital status	
Dead no. (%)	427 (69.2)
Alive no. (%)	151 (24.5)
NOS no. (%)	39 (6.3)
Average FU (months)	14.7
PTEN Mutations no. pts (%)	137 (22.2)

FU: Follow-up; IDH: Isocitric Dehydrogenase; no.: Number; NOS: Not Otherwise Specified; PTEN: phosphatase and tensin homolog. SD: standard deviation; yrs: years.

## Data Availability

All data are included in the main text.

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
