# Peer review of "Immune Landscape in PTEN-Related Glioma Microenvironment: A Bioinformatic Analysis"

_brainsci, 2022, doi:10.3390/brainsci12040501_

Round 1

Reviewer 1 Report

Authors present an observational study on Phosphatase and tensin homolog (PTEN) gene expression profile as well as immune infiltrates in the microenvironment in high grade gliomas as a prognostic factor for survival, with the data extracted from The Cancer Genome Atlas and genetic data from the GTEx online database. PTEN was found mutated in 22.2% of the 617 TCGA-GBMs patients,  survival curves revealed a worse OS and DFS, albeit not significant, for the high-PTEN profile GBMs. Spearman’s analysis of immune cells demonstrated a strong positive correlation between the PTEN status and infiltration of Treg  and M2 macrophages and the half-maximal inhibitor concentration of TMZ was proved to be lower for PTEN-mutated GBMs compared with PTEN-wild types.

Introduction is well written and provides sufficient information. Materials and Methods, especially the data extraction, need to be revised by a molecular biologist/neurobiologist. Discussion is too short and needs to broadened up.

Several remarks:

Drawback of the study is low number of patients and short follow up, as 62% of patients were still alive at medium 14,6 months of follow up.

What was the ratio of IDH-wildtyp and other GBMs in the interrogated cohort?

Zhou F, Shi Q, Fan X, Yu R, Wu Z, Wang B, Tian W, Yu T, Pan M, You Y, Wang Y. Diverse Macrophages Constituted the Glioma Microenvironment and Influenced by PTEN Status. Front Immunol. 2022 Feb 21;13:841404. doi: 10.3389/fimmu.2022.841404. PMID: 35265085; PMCID: PMC8899089. - this group found high expression of PTEN was associated with a more extended survival period of glioma patients, positively correlated with M2 macrophages and negatively with M1 macrophages and that PTEN status could prevent M1 to M2 polarization and M2 macrophage recruitment of gliomas in vitro - this is contrary to your results; how do you explain this discrepancy-please comment. Is it possilbe to interrogate this effect on your cohort; this would enrich your results and shed a possible light on a somewhat positive influence of PTEN;

I suggest to add to Discussion and comment:

Erira A, Velandia F, Penagos J, Zubieta C, Arboleda G. Differential Regulation of the EGFR/PI3K/AKT/PTEN Pathway between Low- and High-Grade Gliomas. Brain Sci. 2021 Dec 18;11(12):1655. doi: 10.3390/brainsci11121655. PMID: 34942957; PMCID: PMC8699139.

Zhang P, Meng X, Liu L, Li S, Li Y, Ali S, Li S, Xiong J, Liu X, Li S, Xia Q, Dong L. Identification of the Prognostic Signatures of Glioma With Different PTEN Status. Front Oncol. 2021 Jul 14;11:633357. doi: 10.3389/fonc.2021.633357. PMID: 34336645; PMCID: PMC8317988.

Author Response

Immune Landscape in PTEN-related Glioma Microenvironment: A Bioinformatic Analysis

Response to Reviewer

We want to thank the kind Reviewer for the comments and suggestions that have been precious for us to improve the quality and clarity of our manuscript.

We have made substantial revisions to our manuscript and, below, we have reported an itemized, point-by-point response to the Reviewer remarks.

All the changes in the text have been reported in track change mode ON.

Reviewer #1

  • “Authors present an observational study on Phosphatase and tensin homolog (PTEN) gene expression profile as well as immune infiltrates in the microenvironment in high grade gliomas as a prognostic factor for survival, with the data extracted from The Cancer Genome Atlas and genetic data from the GTEx online database. PTEN was found mutated in 22.2% of the 617 TCGA-GBMs patients, survival curves revealed a worse OS and DFS, albeit not significant, for the high-PTEN profile GBMs. Spearman’s analysis of immune cells demonstrated a strong positive correlation between the PTEN status and infiltration of Treg and M2 macrophages and the half-maximal inhibitor concentration of TMZ was proved to be lower for PTEN-mutated GBMs compared with PTEN-wild types.

Introduction is well written and provides sufficient information. Materials and Methods, especially the data extraction, need to be revised by a molecular biologist/neurobiologist. Discussion is too short and needs to broadened up.”

We want to thank the kind Reviewer for these comments.

The Materials and Methods section was edited with the support of a computer engineer and a neurobiologist, who were fundamental for the software interpretation, data extraction, and statistical analysis.

Moreover, we agree with the need to improve the discussion section and, according to the reviewer’s suggestion, we have entirely rearranged and expanded the discussion as follows: “The present study analyzes mutation profiles and immune signatures of PTEN-associated microenvironment estimating GBM patients' prognosis, survival, and chemotherapy response. PTEN gene regulates the cell cycle and DNA repair mechanisms. Its expression modulates cell proliferation, neural development, and gliogenesis [21,27,42].

PTEN mutations are hallmarks of glioma malignancy and influence the patients’ survival [43-49]. As recently reported by Erira and his group, alterations of PTEN genes are related to glioma proliferation. PTEN mutation may induce post-translational changes in low-grade gliomas, leading to malignant progression [50].

The prognostic role of the PTEN status has been widely deepened in the literature. In 2001, Sasaki et al. analyzed the different median survival of glioma patients as it related to the PTEN expression. They reported a better OS for wild-type PTEN gliomas (123.4 months), compared to the mutated ones (14.8 months) [51]. In 2016, Han and colleagues explored the genetic linkage between PTEN expression and patients’ outcomes in a meta-analysis where a worse prognosis was revealed for PTEN-mutated gliomas [49]. Zhang and his group, 2021, conducted an online bioinformatics analysis about PTEN mutation as a prognostic signature for HGGs. They designed a tailored risk score based on the individual PTEN status, aiming to simplify HGGs diagnosis, prognosis, and treatment planning. They identified 14 independent prognostic genes in PTEN-wild type tumors and 3 for the PTEN-mutated ones. These last proved to be related to the worse survival [52].

Even without any statistical significance, our Kaplan–Meier analysis confirmed dismal OS (Log-rank p= 0.58, HR= 0.91, p=0.6) and DFS (Log-rank p= 0.78, HR= 1, p=0.82) for the high-PTEN mutation profiles.

Amid the PTEN-related genetic mechanisms underlying the tumorigenesis, the maintenance of the glioma immune microenvironment is critical. Based on this rationale, our analysis also aimed at typifying the subpopulations at the glioma immune niche.

Our Spearman’s correlation tests of immune subpopulations reported a negative correlation (ρ<0) between PTEN mutations and the expression of infiltrating T cell CD4+ Th 1 (ρ = -0.245), Th 2 (ρ = -0.225), and NK cells (ρ = -0.163) within the tumor microenvironment. The GEPIA one-way ANOVA analysis also revealed a lower density of B cell naive (p< 1e -15), CD8+ (p=0.08), CD4+ naive (p< 1e -15), and NK cell (p=3.97e -9) concomitant with TCGA-GBM high-PTEN mutation profiles, compared to the GTEx samples.

In accordance with literature evidence, these data denoted brain tumors growth and progression are sustained by genetic mechanisms of immune tolerance and exhaustion. The suppression of T, B, and NK cells activity within the PTEN tumor microenvironment suggested immune-mediated biological processes are involved in pathways for glioma immune evasion and resistance to chemotherapies [53-56].

On the contrary, through Spearman’s analysis the Treg (ρ= 0.179), endothelial cells (ρ= 0.3.03), and monocyte/macrophages (ρ= 0.368) were found prevalent. The differential examination demonstrated the polarization of M2 (ρ= 0.303) within the glioma microenvironment, versus the monocytes (ρ= -0.205). The GEPIA Treg (p= 0.07) and macrophages (p= 3.49e -3) were the most represented subtypes, with a prevalence of M2 in the TCGA-GBM group.

Tregs preserve immune homeostasis, contribute to the downregulation of T cell activity, and regulate innate and adaptive responses against self-antigens, allergens, and infectious agents [57-61]. Tregs also act as immunosuppressive within the tumor microenvironment, repressing the function of CD4+, CD8+, and NK cells [62]. The immune control is carried out by cytokines, extracellular vesicles, perforins, and cytolytic enzymes [63]. They repress antitumor immunity and facilitate immune escape mechanisms, resulting in glioma progression and relapse [64-66].

Our results also verified that the immunosuppressive anticancer microenvironment is sustained by the recruitment of monocytes which in the glioma context are converted into macrophages, with an explicit M2 polarization. M2 macrophages are known to hold an immunosuppressive role [67,68]. M2 phenotypes induce the differential expression of receptors, cytokines, and chemokines. They produce IL-10, IL-1, and IL-6, stimulating tumorigenesis and negatively affecting the prognosis [69,70]. The M2 macrophages, detected in perivascular areas, enhance the VEGF and COX2 production resulting in increased and aberrant angiogenesis [71-73].

Therefore, Treg and M2 cells stimulate the glioma cell proliferation, invasion, and support immune escape mechanisms [74-77]. These data were confirmed by our bioinformatic analysis (ρ=0.3.03).

Opposing our results, the latest study by Zhou and colleagues published in 2022, found the high expression of PTEN related to a better prognosis for HGGs patients [78]. They discovered via the transwell and flow cytometry that the PTEN gene may inhibit the M1/M2 polarization and M2 macrophages recruitment. These data suggest a potential positive role of the PTEN as an antitumoral immunoregulatory gene [78].

The discrepancy from our study can be explained by the distinct patients’ cohorts involved, as the Chinese Glioma Genome Atlas (CGGA) database, and the different techniques applied for data analysis. But, above all, they explored the effects of different PTEN statuses, splitted into PTEN deletion, PTEN mutated, and PTEN wild type. This distinction allowed to assess the specific impact of each PTEN gene expression on glioma immunity. The study demonstrates that, despite advances in genomics, further researches are needed to shed light on PTEN activity and its immunological role in tumor progression.

The composition of immune infiltrates explains the prognosis concomitant to the PTEN status and, above all, the ineffective response to standard therapies. TMZ is currently the first-line treatment for HGGs in combination with surgery and radiotherapy [79,80]. It is still debated whether the PTEN mutation may influence sensitivity to radiochemotherapy [81-83]. PTEN controls the Wnt/β-catenin and PI3K/Akt/mTOR signaling pathways and arrests the cell cycle at the G2/M phase. TMZ alkylates DNA at this stage, hence PTEN overexpression affects the complex biochemical mechanism of drug alkylation encouraging TMZ activity [84,85].

In 2012 Carico and colleagues conducted a clinical study involving newly diagnosed GBM treated with TMZ. They reported greater effectiveness of TMZ in GBMs with PTEN loss. Inaba et al. investigated the effects of TMZ related to PTEN status founding an increased efficacy in cases of PTEN mutation [83]. Similarly, our analysis of the GDSC database revealed a lower IC50 for PTEN-mutated GBMs (531.13 μM) in comparison with wild-types (701.55 μM).

Apart from conventional chemotherapy, the identification of immune phenotypes and molecular interactions within the tumor microenvironment has been recognized as crucial to widening the spectrum of tailored strategies against the immune escape mechanisms [86-88].

Limitations of the Study

The present study has several undeniable limitations, among which the relatively limited number of patients and short follow-up (average 14,7 months). Other potential biases were the different patients’ ethnicity, limited data about the histological classification, and heterogeneity of radiochemotherapy regimens.” (Pages 11-13, Lines 207-351)

  • “Several remarks: Drawback of the study is low number of patients and short follow up, as 62% of patients were still alive at medium 14,6 months of follow up.”

We agree with the Reviewer that these aspects constitute the major limits of our study. Following this suggestion, we rearranged the "Limitations of the Study" section as follows: “Limitations of the Study

The present study has several undeniable limitations, among which the relatively limited number of patients and short follow-up (average 14,7 months). Other potential biases were the different patients’ ethnicity, limited data about the histological classification, and heterogeneity of radiochemotherapy regimens.” (Page 13, Lines 347-351)

  • “What was the ratio of IDH-wildtype and other GBMs in the interrogated cohort?

Thank you for this suggestion, based on which we had the opportunity to integrate this pivotal information into our manuscript, introducing the following amendments:

- the ratio of IDH-1 wild-type and IDH1-mutant GBMs was added in Table 1;

- the IDH percentage of mutation was included in Figure 1;

- the results section was rearranged as follows: " PTEN was the most frequent gene by the transcriptome examination (22.2%). It was followed by TTN (20.75%), TP53 (20.10%), EGFR (17.18%), FLG (12.64%), MUC16 (11.51%), NF1 (8.27%), RYR2 (7.62%), PKHD1 (7.29%), HMCN1 (7.29%), SYNE1 (7.29%), SPTA1 (6.97%), PIK3R1 (6.97%), RB1 (6.81%), ATRX (6.65%), IDH (6.62%), PIK3CA (6.48%), OBSCN (6.48%), APOB (6.32%), FLG2 (6.32%), and LRP2 (6.16%) (Figure 1)." (Page 4, Lines 139-143)

  • “Zhou F, Shi Q, Fan X, Yu R, Wu Z, Wang B, Tian W, Yu T, Pan M, You Y, Wang Y. Diverse Macrophages Constituted the Glioma Microenvironment and Influenced by PTEN Status. Front Immunol. 2022 Feb 21;13:841404. doi: 10.3389/fimmu.2022.841404. PMID: 35265085; PMCID: PMC8899089. –

this group found high expression of PTEN was associated with a more extended survival period of glioma patients, positively correlated with M2 macrophages and negatively with M1 macrophages and that PTEN status could prevent M1 to M2 polarization and M2 macrophage recruitment of gliomas in vitro - this is contrary to your results; how do you explain this discrepancy-please comment. Is it possible to interrogate this effect on your cohort; this would enrich your results and shed a possible light on a somewhat positive influence of PTEN”

We are particularly grateful to the kind Reviewer for this paramount suggestion, in the light of which we have modified the discussion with the aim to enrich our analysis and explain the discrepancy that emerged from the study suggested. We added the following period: “Opposing our results, the latest study by Zhou and colleagues published in 2022, found the high expression of PTEN related to a better prognosis for HGGs patients [78]. They discovered via the transwell and flow cytometry that the PTEN gene may inhibit the M1/M2 polarization and M2 macrophages recruitment. These data suggest a potential positive role of the PTEN as an antitumoral immunoregulatory gene [78].

The discrepancy from our study can be explained by the distinct patients’ cohorts involved, as the Chinese Glioma Genome Atlas (CGGA) database, and the different techniques applied for data analysis. But, above all, they explored the effects of different PTEN statuses, splitted into PTEN deletion, PTEN mutated, and PTEN wild type. This distinction allowed to assess the specific impact of each PTEN gene expression on glioma immunity. The study demonstrates that, despite advances in genomics, further researches are needed to shed light on PTEN activity and its immunological role in tumor progression.”(Page 12, Lines 275-287)

  • “I suggest to add to Discussion and comment:

Erira A, Velandia F, Penagos J, Zubieta C, Arboleda G. Differential Regulation of the EGFR/PI3K/AKT/PTEN Pathway between Low- and High-Grade Gliomas. Brain Sci. 2021 Dec 18;11(12):1655. doi: 10.3390/brainsci11121655. PMID: 34942957; PMCID: PMC8699139.

Zhang P, Meng X, Liu L, Li S, Li Y, Ali S, Li S, Xiong J, Liu X, Li S, Xia Q, Dong L. Identification of the Prognostic Signatures of Glioma With Different PTEN Status. Front Oncol. 2021 Jul 14;11:633357. doi: 10.3389/fonc.2021.633357. PMID: 34336645; PMCID: PMC8317988.”

We want to thank the kind Reviewer for this advice, based on which we modified the discussion by adding the following sentences:

  • PTEN mutations are hallmarks of glioma malignancy and influence the patients’ survival [43-49]. As recently reported by Erira and his group, alterations of PTEN genes are related to glioma proliferation. PTEN mutation may induce post-translational changes in low-grade gliomas, leading to malignant progression [50].” (Page 11, Lines 211-214)
  • Zhang and his group, 2021, conducted an online bioinformatics analysis about PTEN mutation as a prognostic signature for HGGs. They designed a tailored risk score based on the individual PTEN status, aiming to simplify HGGs diagnosis, prognosis, and treatment planning. They identified 14 independent prognostic genes in PTEN-wild type tumors and 3 for the PTEN-mutated ones. These last proved to be related to the worse survival [52].”(Page 11, Lines 220-225)

The recommended articles were included in the references section.

We want to thank once again the kind Reviewer for the valuable suggestions which have been paramount for us to improve the overall clarity and quality of the manuscript.  

The Authors

Reviewer 2 Report

1. In the introduction, please mention that "the current standard of care in the management of high grade glioma is maximum surgical resection, adjuvant chemoradiation, and six cycles of temozolomide" and use this paper as a reference [before this sentence Despite advances 40 in surgical techniques, diagnostics, and target therapeutic strategies, the 5-year survival 41 rate persists under 10% ......]. 

A Systematic Review and Meta-Analysis on the Number of Adjuvant Temozolomide Cycles in Newly Diagnosed Glioblastoma https://doi.org/10.3389/fonc.2021.779491

2. In the introduction, please appreciate current works on the classification of glioma based on Genes, Molecules, Pathways, and/or Combinations. Please read this paper for more data [before this sentence Phosphatase and tensin homolog (PTEN), a tumor].

The 2021 WHO Classification of Tumors of the Central Nervous System: a summary https://doi.org/10.1093/neuonc/noab106

3. PTEN pathway has a close interaction with Wnt/β-catenin pathway and PI3K/AKT/mTOR Signaling pathway that should be described in the introduction properly. Please read and cite these papers in this context.  

Therapeutic potential of targeting the Wnt/β-catenin pathway in the treatment of pancreatic cancer https://doi.org/10.1002/jcb.27835

The Esophageal Cancer and the PI3K/AKT/mTOR Signaling Regulatory microRNAs: a Novel Marker for Prognosis, and a Possible Target for Immunotherapy https://doi.org/10.2174/1381612825666190110143258

4. In results, please split data of patients with grade III and IV glioma.

Author Response

Immune Landscape in PTEN-related Glioma Microenvironment: A Bioinformatic Analysis

Response to Reviewer

We want to thank the kind Reviewer for the comments and suggestions that have been precious for us to improve the quality and clarity of our manuscript.

We have made substantial revisions to our manuscript and, below, we have reported an itemized, point-by-point response to the Reviewer remarks.

All the changes in the text have been reported in track change mode ON.

Reviewer #2

  • “1. In the introduction, please mention that "the current standard of care in the management of high-grade glioma is maximum surgical resection, adjuvant chemoradiation, and six cycles of temozolomide" and use this paper as a reference [before this sentence Despite advances 40 in surgical techniques, diagnostics, and target therapeutic strategies, the 5-year survival 41 rate persists under 10% ......].

A Systematic Review and Meta-Analysis on the Number of Adjuvant Temozolomide Cycles in Newly Diagnosed Glioblastoma https://doi.org/10.3389/fonc.2021.779491

We want to thank the kind Reviewer for this precious suggestion, according to which we added the following sentence in the introduction section: “The current standard of care in the management of HGGs is maximum surgical resection, adjuvant chemoradiation, and six cycles of temozolomide [8].” (Page 1, Lines 40-43)

The suggested article was included in the references.

  • “2. In the introduction, please appreciate current works on the classification of glioma based on Genes, Molecules, Pathways, and/or Combinations. Please read this paper for more data [before this sentence Phosphatase and tensin homolog (PTEN), a tumor].

The 2021 WHO Classification of Tumors of the Central Nervous System: a summary https://doi.org/10.1093/neuonc/noab106

We are particularly grateful to the Reviewer for this paramount advice, thanks to which we included the following sentence in the introduction: “In 2021, Louis and colleagues published the 5th edition of the WHO classification of brain cancers, which reflects the advances in translational medicine, taxonomy, and genetics in neuro-oncology.

They reported a novel tumors nomenclature aimed at integrating histological features, key diagnostic genes, and molecular characteristics underlying oncogenesis [18].” (Page 2, Lines 48-52)

The recommended reference was also reported.

  • “3. PTEN pathway has a close interaction with Wnt/β-catenin pathway and PI3K/AKT/mTOR Signaling pathway that should be described in the introduction properly. Please read and cite these papers in this context.

Therapeutic potential of targeting the Wnt/β-catenin pathway in the treatment of pancreatic cancer https://doi.org/10.1002/jcb.27835

The Esophageal Cancer and the PI3K/AKT/mTOR Signaling Regulatory microRNAs: a Novel Marker for Prognosis, and a Possible Target for Immunotherapy https://doi.org/10.2174/1381612825666190110143258

In the light of this recommendation, we rearranged the introduction section as follows: “Phosphatase and tensin homolog (PTEN), a tumor suppressor gene, is closely involved in cell translation, proliferation, and tumorigenesis [19-22]. PTEN protein blocks the intracellular pathways of phosphatidylinositol 3-kinase/AKT/mammalian target of rapamycin (PI3K/AKT/mTOR) via dephosphorylation of phosphatidylinositol-3-triphosphate, resulting in inhibition of the cell cycle [23-26].

Moreover, the PTEN pathway has a close interaction with the Wnt/β-catenin signals involved in embryogenesis and the determination of neural patterning [22].

Lack/mutation of the PTEN gene, found in 40% of GBMs, influences neurogenesis, gliogenesis and heightens the DNA damage repairing and the malignant progression of brain tumors [20,25,27].” (Page 2, Lines 53-61)

The suggested articles were integrated into the reference section.

  • “4. In results, please split data of patients with grade III and IV glioma.”

We agree with the propriety of splitting histological grading information, but the data extracted from the TCGA reports only the glioblastoma, namely IV WHO grade.

We want to thank once again the kind Reviewer for the valuable suggestions which have been paramount for us to improve the overall clarity and quality of the manuscript.  

We hope that this newly edited version of our manuscript may have improved its quality.

The Authors

Reviewer 3 Report

Minor:

  • Paragraph 195-196 “A ρ<0 of the Spearman test was proved about the expression of infiltrating T cell CD4+ Th 1, Th 2, CD8+, NK cells.”: The discussion is unclear. You have to explain more clearly how the expression of infiltrating T cell CD4+ Th 1, Th 2, CD8+, NK cells are associated with PTEN, glioblastoma, glioma microenvironment. Some studies already reported that the tumor microenvironment immunosuppresses T cell activity and functions. Do you mean that your result proved it?
  • Figure 6C: Here, you missed the result explanation of 6C. Why and what is the meaning of negative correlation of Monocyte and how is it associated with PTEN, glioblastoma, glioma microenvironment? What is different from Monocyte/macrophages versus Monocyte? Some studies reported that the infiltration of monocyte and transformation of Monocytes to macrophages occur in glioma. Does your data match this evidence? You have to explain and discuss your data meaning like your discussion and explanation of resulting data from Paragraph 197, for instance, the downregulation of the T-cell activity by Treg expression, immunosuppressive role of M2 macrophages, etc.
  • Figure 4: I did not read any discussion for the results of Figure 4.
  • Paragraph 211 “Accordingly, PTEN overexpression affects the pharmacokinetics of TMZ.”: I disagree with this point. A drug pharmacokinetics is dependent on tissue/blood partition ratio, drug absorption to penetrate the gut and pass liver metabolism into blood circulation, drug molecule penetration through blood-brain-barrier, organ excretion, such as a kidney. PTEN gene expression will not affect Temozolomide pharmacokinetics. PTEN gene expression tends to affect a complex and complicated biochemical mechanism of anticancer Temozolomide alkylation or affect other immune responses to facilitate Temozolomide alkylation. Temozolomide resistant results from P-glycoprotein effluxing Temozolomide out off glioma, the activity of DNA mismatch repair(MMR) protein to remove Temozolomide alkylation and a high-packing array of glioma to affect Temozolomide molecule distribution into glioma.  Please provide a correct reference to support the paragraph “Accordingly, PTEN overexpression affects the pharmacokinetics of TMZ.”, or you revise this discussion or conclusion. The reference 60 did not report and support this point.

Author Response

Immune Landscape in PTEN-related Glioma Microenvironment: A Bioinformatic Analysis

Response to Reviewer

We want to thank the kind Reviewer for the comments and suggestions that have been precious for us to improve the quality and clarity of our manuscript.

We have made substantial revisions to our manuscript and, below, we have reported an itemized, point-by-point response to the Reviewer remarks.

All the changes in the text have been reported in track change mode ON.

Reviewer #3

  • “Paragraph 195-196 “A ρ<0 of the Spearman test was proved about the expression of infiltrating T cell CD4+ Th 1, Th 2, CD8+, NK cells.”: The discussion is unclear. You have to explain more clearly how the expression of infiltrating T cell CD4+ Th 1, Th 2, CD8+, NK cells are associated with PTEN, glioblastoma, glioma microenvironment. Some studies already reported that the tumor microenvironment immunosuppresses T cell activity and functions. Do you mean that your result proved it?”

We want to thank the kind Reviewer for this precious suggestion. Intending to better clarify this point, we rearranged the discussion as follows: “Our Spearman’s correlation tests of immune subpopulations reported a negative correlation (ρ<0) between PTEN mutations and the expression of infiltrating T cell CD4+ Th 1 (ρ = -0.245), Th 2 (ρ = -0.225), and NK cells (ρ = -0.163) within the tumor microenvironment. The GEPIA one-way ANOVA analysis also revealed a lower density of B cell naive (p< 1e -15), CD8+ (p=0.08), CD4+ naive (p< 1e -15), and NK cell (p=3.97e -9) concomitant with TCGA-GBM high-PTEN mutation profiles, compared to the GTEx samples.

In accordance with literature evidence, these data denoted brain tumors growth and progression are sustained by genetic mechanisms of immune tolerance and exhaustion. The suppression of T, B, and NK cells activity within the PTEN tumor microenvironment suggested immune-mediated biological processes are involved in pathways for glioma immune evasion and resistance to chemotherapies [53-56].” (Page 12, Lines 245-255)

We also expanded the bibliography added the following articles in references:

- Cordell, E.C.; Alghamri, M.S.; Castro, M.G.; Gutmann, D.H. T lymphocytes as dynamic regulators of glioma pathobiology. Neuro Oncol 2022, 10.1093/neuonc/noac055, doi:10.1093/neuonc/noac055.

- Codrici, E.; Popescu, I.D.; Tanase, C.; Enciu, A.M. Friends with Benefits: Chemokines, Glioblastoma-Associated Microglia/Macrophages, and Tumor Microenvironment. Int J Mol Sci 2022, 23, doi:10.3390/ijms23052509.

- DeCordova, S.; Shastri, A.; Tsolaki, A.G.; Yasmin, H.; Klein, L.; Singh, S.K.; Kishore, U. Molecular Heterogeneity and Immunosuppressive Microenvironment in Glioblastoma. Front Immunol 2020, 11, 1402, doi:10.3389/fimmu.2020.01402.

- Ma, Q.; Long, W.; Xing, C.; Chu, J.; Luo, M.; Wang, H.Y.; Liu, Q.; Wang, R.F. Cancer Stem Cells and Immunosuppressive Microenvironment in Glioma. Front Immunol 2018, 9, 2924, doi:10.3389/fimmu.2018.02924.

  • “Figure 6C: Here, you missed the result explanation of 6C. Why and what is the meaning of negative correlation of Monocyte and how is it associated with PTEN, glioblastoma, glioma microenvironment? What is different from Monocyte/macrophages versus Monocyte? Some studies reported that the infiltration of monocyte and transformation of Monocytes to macrophages occur in glioma. Does your data match this evidence?

You have to explain and discuss your data meaning like your discussion and explanation of resulting data from Paragraph 197, for instance, the downregulation of the T-cell activity by Treg expression, immunosuppressive role of M2 macrophages, etc.”

Thank you for this point, based on which we rearranged the discussion to better clarified the immunosuppressive role of monocytes, Treg, and M2 polarization within the tumor microenvironment, as follows: “Tregs preserve immune homeostasis, contribute to the downregulation of T cell activity, and regulate innate and adaptive responses against self-antigens, allergens, and infectious agents [57-61]. Tregs also act as immunosuppressive within the tumor microenvironment, repressing the function of CD4+, CD8+, and NK cells [62]. The immune control is carried out by cytokines, extracellular vesicles, perforins, and cytolytic enzymes [63]. They repress antitumor immunity and facilitate immune escape mechanisms, resulting in glioma progression and relapse [64-66].

Our results also verified that the immunosuppressive anticancer microenvironment is sustained by the recruitment of monocytes which in the glioma context are converted into macrophages, with an explicit M2 polarization. M2 macrophages are known to hold an immunosuppressive role [67,68]. M2 phenotypes induce the differential expression of receptors, cytokines, and chemokines. They produce IL-10, IL-1, and IL-6, stimulating tumorigenesis and negatively affecting the prognosis [69,70]. The M2 macrophages, detected in perivascular areas, enhance the VEGF and COX2 production resulting in increased and aberrant angiogenesis [71-73].

Therefore, Treg and M2 cells stimulate the glioma cell proliferation, invasion, and support immune escape mechanisms [74-77]. These data were confirmed by our bioinformatic analysis (ρ=0.3.03).” (Page 12, Lines 262-279)

We expanded the bibliography by adding the following articles:

- Grover, P.; Goel, P.N.; Greene, M.I. Regulatory T Cells: Regulation of Identity and Function. Frontiers in immunology 2021, 12, 750542-750542, doi:10.3389/fimmu.2021.750542.

- Strauss, L.; Bergmann, C.; Whiteside, T.L. Human circulating CD4+CD25highFoxp3+ regulatory T cells kill autologous CD8+ but not CD4+ responder cells by Fas-mediated apoptosis. J Immunol 2009, 182, 1469-1480, doi:10.4049/jimmunol.182.3.1469.

- Boer, M.C.; Joosten, S.A.; Ottenhoff, T.H. Regulatory T-Cells at the Interface between Human Host and Pathogens in Infectious Diseases and Vaccination. Front Immunol 2015, 6, 217, doi:10.3389/fimmu.2015.00217.

- Sakaguchi, S. Regulatory T cells: key controllers of immunologic self-tolerance. Cell 2000, 101, 455-458, doi:10.1016/s0092-8674(00)80856-9.

- Scheinecker, C.; Göschl, L.; Bonelli, M. Treg cells in health and autoimmune diseases: New insights from single cell analysis. J Autoimmun 2020, 110, 102376, doi:10.1016/j.jaut.2019.102376.

- Okeke, E.B.; Uzonna, J.E. The Pivotal Role of Regulatory T Cells in the Regulation of Innate Immune Cells. Front Immunol 2019, 10, 680, doi:10.3389/fimmu.2019.00680.

- Nishikawa, H.; Sakaguchi, S. Regulatory T cells in tumor immunity. Int J Cancer 2010, 127, 759-767, doi:10.1002/ijc.25429.

- Takeuchi, Y.; Nishikawa, H. Roles of regulatory T cells in cancer immunity. Int Immunol 2016, 28, 401-409, doi:10.1093/intimm/dxw025.

- Kim, J.H.; Kim, B.S.; Lee, S.K. Regulatory T Cells in Tumor Microenvironment and Approach for Anticancer Immunotherapy. Immune Netw 2020, 20, e4, doi:10.4110/in.2020.20.e4.

- Mantovani, A.; Sozzani, S.; Locati, M.; Allavena, P.; Sica, A. Macrophage polarization: tumor-associated macrophages as a paradigm for polarized M2 mononuclear phagocytes. Trends Immunol 2002, 23, 549-555, doi:10.1016/s1471-4906(02)02302-5.

- Yunna, C.; Mengru, H.; Lei, W.; Weidong, C. Macrophage M1/M2 polarization. Eur J Pharmacol 2020, 877, 173090, doi:10.1016/j.ejphar.2020.173090.

  • “Figure 4: I did not read any discussion for the results of Figure 4.”

In the light of this recommendation, we modified the discussion with the following sentence: “Even without any statistical significance, our Kaplan–Meier analysis confirmed dismal OS (Log-rank p= 0.58, HR= 0.91, p=0.6) and DFS (Log-rank p= 0.78, HR= 1, p=0.82) for the high-PTEN mutation profiles.” (Page 11, Lines 231-233)

  • “Paragraph 211 “Accordingly, PTEN overexpression affects the pharmacokinetics of TMZ.”: I disagree with this point. A drug pharmacokinetics is dependent on tissue/blood partition ratio, drug absorption to penetrate the gut and pass liver metabolism into blood circulation, drug molecule penetration through blood-brain-barrier, organ excretion, such as a kidney. PTEN gene expression will not affect Temozolomide pharmacokinetics. PTEN gene expression tends to affect a complex and complicated biochemical mechanism of anticancer Temozolomide alkylation or affect other immune responses to facilitate Temozolomide alkylation. Temozolomide resistant results from P-glycoprotein effluxing Temozolomide out off glioma, the activity of DNA mismatch repair (MMR) protein to remove Temozolomide alkylation and a high-packing array of glioma to affect Temozolomide molecule distribution into glioma. Please provide a correct reference to support the paragraph “Accordingly, PTEN overexpression affects the pharmacokinetics of TMZ.”, or you revise this discussion or conclusion. The reference 60 did not report and support this point.”

We are particularly grateful to the kind Reviewer for this comment, and we apologize for the error. According to this suggestion, we revised this section as follows: “It is still debated whether the PTEN mutation may influence sensitivity to radiochemotherapy [81-83]. PTEN controls the Wnt/β-catenin and PI3K/Akt/mTOR signaling pathways and arrests the cell cycle at the G2/M phase. TMZ alkylates DNA at this stage, hence PTEN overexpression affects the complex biochemical mechanism of drug alkylation encouraging TMZ activity [84,85].

In 2012 Carico and colleagues conducted a clinical study involving newly diagnosed GBM treated with TMZ. They reported greater effectiveness of TMZ in GBMs with PTEN loss. Inaba et al. investigated the effects of TMZ related to PTEN status founding an increased efficacy in cases of PTEN mutation [83]. Similarly, our analysis of the GDSC database revealed a lower IC50 for PTEN-mutated GBMs (531.13 μM) in comparison with wild-types (701.55 μM).”(Pages 12-13, Lines 296-347)

Furthermore, we included the following articles in the reference section:

- Zając, A.; Sumorek-Wiadro, J.; Langner, E.; Wertel, I.; Maciejczyk, A.; Pawlikowska-Pawlęga, B.; Pawelec, J.; Wasiak, M.; Hułas-Stasiak, M.; Bądziul, D., et al. Involvement of PI3K Pathway in Glioma Cell Resistance to Temozolomide Treatment. Int J Mol Sci 2021, 22, doi:10.3390/ijms22105155.

- Carico, C.; Nuño, M.; Mukherjee, D.; Elramsisy, A.; Dantis, J.; Hu, J.; Rudnick, J.; Yu, J.S.; Black, K.L.; Bannykh, S.I., et al. Loss of PTEN is not associated with poor survival in newly diagnosed glioblastoma patients of the temozolomide era. PLoS One 2012, 7, e33684, doi:10.1371/journal.pone.0033684.

We want to thank once again the kind Reviewer for the valuable suggestions which have been paramount for us to improve the overall clarity and quality of the manuscript.  

The Authors

Round 2

Reviewer 1 Report

The authors have sufficiently repplied to the reviewers remarks.